# Using telemedicine to improve early medical abortion at home (UTAH): a randomised controlled trial to compare telemedicine with in-person consultation for early medical abortion

John Joseph Reynolds-Wright ,[1,2] John Norrie ,[3]
Sharon Tracey Cameron [1,2,4]

¹MRC Centre for Reproductive Health, Institute for Regeneration and Repair, University of Edinburgh, Edinburgh, UK
²Chalmers Centre for Sexual Health, NHS Lothian, Edinburgh, UK
³Edinburgh Clinical Trials Unit, Usher Institute, University of Edinburgh, Bioquarter, Edinburgh, UK
⁴Obstetrics and Gynaecology, University of Edinburgh, Edinburgh, UK

**Correspondence to**
John Joseph Reynolds-Wright;
john.reynolds-wright@ed.ac.uk

## ABSTRACT

**Objectives** To compare telephone consultations with in-person consultations for the provision of medical abortion (using mifepristone 200 mg and misoprostol 800 µg). We hypothesised that telemedicine consultations would be non-inferior to in-person consultations with a non-inferiority limit of 3%.

**Design** Randomised controlled trial with 1:1 allocation.

**Setting** Community abortion service housed within an integrated sexual and reproductive health service in Edinburgh, UK.

**Participants** The trial began on 13 January 2020, but was stopped early due to COVID-19; recruitment was suspended on 31 March 2020, and was formally closed on 31 August 2021. A total of 125 participants were randomised, approximately 10% of the total planned, with 63 assigned to telemedicine and 62 to in-person consultation.

**Primary and secondary outcome measures** Primary outcome: efficacy of medical abortion, defined as complete abortion without surgical intervention. Secondary outcomes: satisfaction with consultation type, preparedness, unscheduled contact with care, complication rate, time spent in clinical contact and uptake of long-acting contraception.

**Results** Primary outcome was available for 115 participants (lost-to-follow-up telemedicine=2, in-person=8), secondary outcomes were available for 110 participants (n=5 and n=10 in telemedicine and in-person groups did not complete questionnaires). There were no significant differences between groups in treatment efficacy (telemedicine 57/63 (90.5%), in-person 48/62 (77.4%)). However, non-inferiority was not demonstrated (+3.3% in favour of telemedicine, CI −6.6% to +13.3%, lower than non-inferiority margin). There were no significant differences in most secondary outcomes, however, there was more unscheduled contact with care in the telemedicine group (12 (19%) vs 3 (5%), p=0.01). The overall time spent in clinical contact was statistically significantly lower in the telemedicine group (mean 94 (SD 24) vs 111 (24) min, p=0.0005).

**Conclusions** Telemedicine for medical abortion appeared to be effective, safe and acceptable to women, with

## STRENGTHS AND LIMITATIONS OF THIS STUDY

⇒ To the best of our knowledge, this is the first randomised controlled trial in a high-income setting to compare telemedicine with in-person consultation for the delivery of early medical abortion care.
⇒ Most other studies of telemedicine in abortion care have been observational cohort studies and so this trial makes a useful addition to the literature.
⇒ Due to COVID-19, the UTAH (UTAH, Using Telemedicine to improve early medical Abortion at Home) study had to close shortly after recruitment began as telemedicine became the standard of care; as such, only 10% of the target sample was recruited and the study is underpowered for outcome assessment.

less time spent in the clinic. However, due to the small sample size resulting from early cessation, the study was underpowered to confirm this conclusion. These findings warrant further investigation in larger scale studies.

**Trial registration number** NCT04139382.

## INTRODUCTION

In-person consultations prior to abortion can be lengthy. Women can struggle with arranging childcare or time off work to attend appointments. There is evidence from observational studies that telemedicine consultations for those seeking early medical abortion (EMA) at under 10 weeks gestation may be a safe and acceptable alternative.[1–5] Additionally, the WHO recommend that women can reliably self-manage much of EMA with support from a healthcare worker.[6]

Prior to COVID-19 in Scotland, women who chose an EMA typically made a single visit to a clinic for a consultation, assessment of gestation, receipt of mifepristone (to be administered in clinic, as per UK legal requirements at the time) and a medication pack to take

home. This contained misoprostol (to self-administer at home contraception where desired, and instructions on how to self-assess the success of the abortion using a self-performed low-sensitivity urinary pregnancy test (LSUPT, detection limit 1000 IU human chorionic gonadotropin) at home 2 weeks later.[7–9]

This clinic visit could last up to 2 hours; much of which may be time spent in the waiting room between seeing clinicians and having an ultrasound to confirm gestation. However, a significant proportion of consultation time is standardised history taking and information giving, and could be conducted via telemedicine by telephone, video call or over the Internet, rather than in-person.

Telemedicine consultations could add flexibility for women (eg, consultation from home), and benefit services with reduced footfall in clinics and allow for more flexible staff working (including remote working). It is also possible that women may be better prepared for EMA: having a telemedicine consultation in advance of receipt of medications may mean they have had more time to process the information provided about EMA, as well as any information about postabortion contraception options. Doing this prior to attending clinic for ultrasound (which was nearly universal prior to COVID-19) and treatment provision, may give further opportunity to clarify areas of uncertainty for abortion treatment or select a contraceptive method.

We therefore aimed to compare telemedicine with in-person consultations regarding effectiveness, safety, acceptability and duration of consultation.

## METHODS
### Study design
The full study protocol has been published.[10] In brief, we designed a randomised controlled trial to compare telemedicine consultations, specifically a telephone call, with in-person consultations for women seeking EMA.

### Participants
Participants were pregnant individuals seeking medical abortion at home, aged 16 or over, able to communicate in English and fulfil the requirements of the 1967 Abortion Act. In addition to being eligible to participate in the trial, women needed to be clinically eligible to receive EMA, which for the study was defined as having a viable intrauterine pregnancy determined by transabdominal or transvaginal ultrasound at or less than 9 weeks and 6 days' gestation.

### Setting
The study was conducted at a single centre in the UK (Chalmers Centre, Edinburgh) that provides abortion care to a predominately urban population with some women living in more remote and rural areas. Standard care was for women to self-refer to the service by telephone and receive a subsequent in-person clinic appointment.[11] At this appointment, they had a routine preabortion ultrasound scan to determine location and gestation of the pregnancy, followed by a consultation with a doctor or nurse to obtain medical history, discuss the chosen method of abortion and provide written informed consent. Counselling about postabortion contraception was offered routinely to all patients. For medical abortion at home, the first medication (mifepristone 200 mg) was provided orally in the clinic as per UK legal requirements at that time. Women were then provided with a medication pack for home use that included misoprostol 800 µg (sublingual or vaginal as per patient preference), contraceptive supplies if desired and LSUPT and written information were provided to take home, which included instructions for using medications, what to expect in terms of pain and bleeding and when to seek medical attention, including calling the clinic for further advice[9 12] (see online supplemental figure 1).

### Randomisation
In the study, at the point of self-referral for abortion, the clinic administrative team asked women if they would be happy to be contacted by a research nurse. Those agreeing received a telephone call the same or next day, during which the clinical research nurses explained the study, obtained consent and randomised women using Research Electronic Data Capture (REDCap) software, hosted by University of Edinburgh.[13 14] The randomisation list was produced by an independent statistician. Due to the nature of the interventions, blinding was not possible.

### Study groups
The 'in-person' group then went on to receive standard care as described above. Following this in-person consultation, they completed a short interviewer-administered questionnaire with the research nurse in clinic.

The 'telemedicine' group received their telephone consultation using the same proforma as for in-person consultations, including discussion of postabortion contraception. This was conducted by the research nurses, who has experience working in the abortion clinic and providing counselling in in-person consultations. This was done either immediately after randomisation or at another time, as preferred by the woman. Following this consultation, the woman then attended the clinic for ultrasound scan, medication provision (mifepristone was administered at the appointment under clinician supervision as per legislation at the time) and completed the study questionnaire (as for in-person care group). Women in both groups received a follow-up telephone call at 2 weeks after EMA from a research nurse to determine the LSUPT result and to administer the follow-up questionnaire. These questionnaire data were combined with routinely collected clinical data on demographics, gestation as determined by preabortion ultrasound reproductive history and deprivation score (Scottish Index of Multiple Deprivation, SIMD[15]). See online supplemental figure 2 for overview of study flow.

## Outcomes

The primary outcome of this study was to determine if EMA conducted following a telemedicine consultation is as effective (complete abortion rate without surgical intervention) as following in-person consultation. Effectiveness was chosen in accordance with the Medical Abortion Reporting of Efficacy (MARE) core outcome recommendations.[16] This was determined in the first instance using self-reported LSUPT result 2 weeks after EMA, any woman with a positive LSUPT result at this point was invited to attend for follow-up ultrasound to confirm/exclude continuing pregnancy. Any woman with a positive LSUPT but whose postabortion ultrasound showed an empty uterus was classed as having a complete abortion. The clinical database was reviewed at 6 weeks post-EMA to check for any admission or surgical intervention to a hospital within the region.

The secondary outcomes were total time spent in clinical contact to receive EMA (calculated as time spent in the clinic plus for the telemedicine group this also included the duration of the telephone consultation), preparedness for EMA as assessed by preabortion questionnaire (published as supplemental file with the open access study protocol[10]), level of satisfaction with consultation as assessed by follow-up questionnaire at 2 weeks (published as supplemental file with the open access study protocol[10]), rates of unscheduled contact with abortion service (initiated by the patient or emergency attendance) and any complications resulting from abortion treatment (by case note review at 6 weeks post-EMA), proportion of women who were found to be ineligible for EMA when attending clinic for preabortion ultrasound.

## Statistical methods

The recruitment target of 1222 was calculated for the primary outcome, success of EMA, using a binary outcome non-inferiority design assuming an EMA success rate of 97% in both telemedicine and in-person groups, with 90% power, one-sided 5% level of significance, 3% non-inferiority limit, 1:1 allocation and around 10% compensation for loss to follow-up.[17] Primarily due to the COVID-19 pandemic, the study only recruited 125 participants, and the observed EMA rate in the standard of care group was lower at 90% rather than the assumed 97%. Indicatively, a study with 125 evaluable participants and assumed EMA 90% could only detect a non-inferiority margin of almost 16% at 90% power and one-sided 5% level of significance.

Descriptive statistics were used to characterise participants and assess comparability of the two groups at baseline.

The primary outcome of complete abortion was compared between the randomised groups (telemedicine vs in-person) by calculating the risk difference and its exact 95% (risk difference calculator, neoweb.org.uk).

Secondary outcomes were analysed using appropriate tests depending on distribution of the data: for continuous data, we have used two sample t-tests and Mann

Whitney tests; for categorical data, we have used $\chi^2$ and Fisher Exact tests. We have not imputed any missing data for the secondary outcomes. Results will be considered statistically significant if p value<0.05. There has been no adjustment for multiple comparisons.

## Early cessation

The study opened to recruitment on 13 January 2020, and was suspended on 31 March 2020, in response to COVID-19 pandemic, as telemedicine became standard care. The Scottish Government subsequently conducted a public consultation to determine whether there was public support for making the COVID-19 legislative changes permanent.[18] The public consultation concluded that the legislative changes allowing telemedicine should continue. This meant that there would not be a return to the previous in-person model of care that the UTAH (UTAH, Using Telemedicine to improve early medical Abortion at Home) study was designed around and so the study was formally closed on 31 August 2021. The decision to suspend and then formally close the study was made by the study team following consultation with research and development and service leads.

## Patient and public involvement

We consulted Abortion Rights Edinburgh, a local abortion and women's rights activism group. They kindly provided feedback on the trial rationale, study design and study protocol prior to submission for ethical approval. They have agreed to disseminate the trial findings to their membership and via their networks.

## RESULTS

During the study period, 575 people had abortions at the study centre. Of these, 128 met inclusion criteria were approached; of these, 125 women were recruited, with 63 (50.4%) randomised to the telemedicine group and 62 (49.6%) randomised to the in-person group. Treatment outcome data were available for 115 women (92%); 15 women (12%) did not complete follow-up questionnaires (telemedicine=5, in-person=10). Those who responded to follow-up contacts confirmed that they had used medications as directed with a mifepristone–misoprostol interval of 24–48 hours; however, the specific interval per participant was not collected.

Demographics are summarised in table 1, there were no substantial differences between groups except for SIMD—the in-person group had a higher proportion of women from more deprived backgrounds.

## Primary outcome

The complete abortion rate was high and not statistically significantly different between the two groups with a greater degree of loss to follow-up in the in-person group (table 2). The risk difference of failed abortion (continuing pregnancy) was 3.2% (CI −7.6 to −1.2).

### Table 1 Demographics

| | Telemedicine | In-person |
|---|---|---|
| BMI kg/m$^2$/mean | 26.4 (n=56) | 26.6 (n=52) |
| Parity—total mean | 1.2 (n=63) | 1.6 (n=62) |
| Parity—live births (mean) | 0.6 (n=63) | 0.98 (n=62) |
| Age (mean) | 28.2 (n=63) | 27.5 (n=62) |
| Smoker (n/total, %) | 14/63, 22.2 | 8/62, 12.9 |
| Gestational age weeks+days (median) | 6+6 | 6+5 |
| Gestational age weeks+days (range) | Less than 5 to 9+6 | Less than 5–9+2 |
| Previous live birth (%) | 23 (36.5) | 29 (46.8) |
| Previous abortion (%) | 19 (30.2) | 15 (24.2) |
| SIMD | | |
| Unknown | 1 | 1 |
| 1 (most deprived) | 5 | 18 |
| 2 | 11 | 13 |
| 3 | 11 | 11 |
| 4 | 16 | 5 |
| 5 (least deprived) | 19 | 14 |

BMI, body mass index; SIMD, Scottish Index of Multiple Deprivation.

Since only around 10% of the target sample of 1222 was recruited, the study was generally substantially under powered for this outcome.

### Secondary outcomes

There were no complications resulting from EMA in either the in-person or telemedicine group. No patients in either group required hospitalisation, blood transfusion, emergency department visits and there were no deaths. No patients were found to be ineligible for EMA

### Table 2 Treatment outcome

| | Telemedicine, N (%) | In-person, N (%) |
|---|---|---|
| Complete abortion (negative LSUPT or empty uterus on post-treatment scan) | Of all enrolled: 57/63 (90.5) Excluding loss to follow-up: 57/61 (93.4) | Of all enrolled: 48/62 (77.4) Excluding loss to follow-up: 48/54 (88.9) |
| Continuing pregnancy after treatment | 0 | 2/62 (3.2) |
| Miscarriage prior to treatment | 1 (1.6) | 3 (4.8) |
| Not pregnant at clinic attendance | 2 (3.2) | 1 (1.6) |
| Ectopic on ultrasound | 1 (1.6) | 0 |
| Lost to follow-up | 2/63 (3.2) | 8/62 (12.9) |

LSUPT, low-sensitivity urinary pregnancy test.

### Table 3 Preparedness for treatment and satisfaction with treatment

| | Telemedicine, N (%) | In-person, N (%) |
|---|---|---|
| Preparedness | | |
| Very prepared | 51 (81.0) | 44 (71.0) |
| Somewhat prepared | 6 (9.5) | 8 (12.9) |
| Neutral | 1 (1.2) | 0 |
| Somewhat unprepared | 0 | 0 |
| Very unprepared | 0 | 0 |
| Missing | 5 (7.9) | 10 (15.9) |
| Satisfaction | | |
| Very satisfied | 56 (88.9) | 51 (82.3) |
| Somewhat satisfied | 2 (3.2) | 1 (1.6) |
| Neutral | 0 | 0 |
| Somewhat unsatisfied | 0 | 0 |
| Very unsatisfied | 0 | 0 |
| Missing | 5 (7.9) | 10 (15.9) |

in the telemedicine group when they attended for ultrasound (ie, gestation was not higher than expected). A higher proportion of the telemedicine group reported a positive LSUPT and had a planned further attendance for repeat ultrasound to confirm complete abortion (telemedicine 12/63, 19% vs in-person 3/62, 4.8%), however this outcome was underpowered.

Following the consultation, similar high proportions of women rated themselves as very prepared and very satisfied with the mode of consultation they received for their abortion care. See table 3.

There was no statistically significant difference in the types of postabortion contraception selected between groups when comparing long-acting reversible methods with all other methods; nor with short acting methods, condoms and no method; nor more effective methods (hormonal and intrauterine) with less effective methods. See table 4.

The mean total time spent in clinical contact with the service (telemedicine consultation and clinic visit combined for the telemedicine group; clinic visit time for the in-person group) was significantly lower (p<0.001) in the telemedicine group compared with the in-person group. See table 5.

### DISCUSSION

Due to COVID-19, the UTAH study had to shortly after recruitment began as telemedicine became the standard of care and randomising women to in-person appointments was deemed unsafe for COVID-19 transmission. As a result, the study outcomes are underpowered. However, the findings that effectiveness of treatment, satisfaction and preparedness are all similar between groups reflect data from observational studies about the safety,

**Table 4** Postabortion contraception

|  | Telemedicine N (%) | In-person N (%) |
|---|---|---|
| Copper IUD | 2 (3.2) | 1 (1.6) |
| Hormonal IUD | 3 (4.8) | 2 (3.2) |
| Subdermal implant | 2 (3.2) | 6 (9.7) |
| Injectable | 6 (9.5) | 0 |
| Combined pill | 11 (17.5) | 11 (17.7) |
| Combined patch | 0 | 1 (1.6) |
| Progestogen only pill | 9 (14.3) | 19 (30.6) |
| Condoms | 14 (22.2) | 8 (12.9) |
| No method | 16 (25.4) | 14 (22.6) |

IUD, intrauterine device .

effectiveness and acceptability of telemedicine abortion care.

The only adequately powered outcome was that time spent in preabortion clinical contact was significantly lower in the telemedicine group than in the in-person group. In a service utilising the model of telephone consultation followed by an ultrasound scan for gestational age and medication provision, the total clinical contact time may be even lower in practice still as in this study there were sometimes delays if telephone group patients got 'stuck in the queue' behind standard care patients in the clinic waiting for their ultrasound scan or medications. Models of care where an ultrasound is not required at all may reduce overall time in clinical care even further, as the ultrasound component lasted a mean of 69 min in this study.

The rate of contact following abortion with a positive LSUPT was higher in the telemedicine group compared with the in-person group, however this outcome was underpowered and so may not have been borne out if the study had completed recruitment. If there was a true difference, this may possibly have resulted from greater familiarity and confidence in contacting the clinic via telephone as this was the main method that these participants had used prior to abortion. While all patients were advised to make contact with the clinic if they had a positive or invalid result, the telemedicine group may have

been more confident in doing so, having received their prior care in this way.

The complete abortion rate in the in-person group (77%) is not statistically different from that of the telemedicine group but appears lower than reported elsewhere in the literature.[19] This is likely to be due to the small numbers included in this study, meaning that this outcome is underpowered. Additionally, the participants who did not respond to follow-up contacts in this study arm did not have continuing pregnancies reported on their electronic patient record and so it is likely that they too had a complete abortion. The electronic patient record covers all the hospitals in the region and due to the geography of the area, patients uncommonly travel to neighbouring health boards.

The SIMD composition of the two study groups were different, with a greater proportion of the in-person group living in more deprived areas. This is likely to be due to chance and the small, under-recruited sample for the study.

In addition to this study, there is now a single RCT[20] in South Africa conducted since the start of COVID-19. In that study, those in the telemedicine arm had abdominal palpation from a trained nurse to assess fundal height and screen for pregnancies greater than 12 weeks' gestation. Last menstrual period was used to include patients with pregnancies less than 9 weeks' gestation. That study demonstrated non-inferiority of telemedicine to standard care with regard to complete abortion rates and a preference among women receiving abortion for telemedicine. Their findings reflect the body of observational literature[2 4 5 21] regarding telemedicine abortion care, including other observational cohort studies in the UK conducted during COVID-19.[22–24] While the findings of the UTAH study were underpowered to confirm or refute non-inferiority, they also reflect these findings of Endler *et al*.

The legislation allowing medical abortion at home across Great Britain is now permanent;[18 25] however, many services in Great Britain and worldwide continue to provide abortion through in-person appointments. Moving assessment consultations to be done by telemedicine may result in time savings even if inviting the majority of patients for subsequent ultrasound.

**Table 5** Time spent in clinical contact

| Total min contact (min) | Telemedicine (teleconsult duration only) n=55 | Telemedicine (in clinic duration only) n=55 | Telemedicine (teleconsult and in clinic combined) n=55 | In-person (time in clinic) n=50 |
|---|---|---|---|---|
| Mean | 24 | 69 | 94 | 111 |
| SD | 6 | 23 | 24 | 24 |
| Median | 25 | 68 | 95 | 108 |
| Range | 10–40 | 25–125 | 55–151* | 60–165 |

*Range of teleconsult and in clinic combined is different from ranges in first two columns as those with shortest/longest teleconsult did not have shortest/longest clinic time.

We theorised that that this model of care may give women more time to consider contraceptive options and make a decision to select a more effective method than when confronted with this choice in a face-to-face clinic. However, in this trial as well as in subsequent observational cohort studies, we have conducted at the same centre during COVID-19,[26 27] this has not appeared to be the case. Further study is needed to better link women to effective postabortion contraception.

This is the first RCT in a high income setting to compare telemedicine with in-person appointments for the delivery of abortion care. The study was designed in line with the Consolidated Standards of Reporting Trials (CONSORT) and Medical Abortion Reporting of Efficacy (MARE) guidance.[16] The majority of all other studies of telemedicine in abortion care have been observational cohort studies,[1–3 5] and so this trial makes a useful addition to the literature. However, as outlined above, the early cessation and underpowering of most study outcomes is the major limitation of the study.

## CONCLUSION

Telemedicine for medical abortion appeared to be effective, safe and acceptable to women, with less time spent in the clinic. However, due to the small sample size resulting from the early cessation of recruitment, the study was underpowered to confirm this conclusion. These findings warrant further investigation in larger scale studies.

**Acknowledgements** The authors would like to thank Karen McCabe and Anne Johnstone, clinical research nurses who helped to recruit participants and conduct the study; and the staff and patients at the Chalmers Choices service.

**Contributors** STC conceived the study. JJR-W, JN and STC designed the study protocol. JJR-W designed the study tools and led data collection. JJR-W, JN and STC reviewed and analysed the data. JJR-W wrote the initial manuscript. JJR-W, JN and STC all contributed to the final manuscript. JJR-W is the guarantor for the manuscript.

**Funding** This work was supported by the Edinburgh Family Planning Trust grant number EFPT/2019/UTAH. The project was conducted from the MRC Centre for Reproductive Health, supported by the Medical Research Council (grant MR/N022556/1).

**Competing interests** JRW has received an educational grant from Gedeon Richter. STC and JN have no competing interests to declare.

**Patient and public involvement** Patients and/or the public were involved in the design, or conduct, or reporting or dissemination plans of this research. Refer to the Methods section for further details.

**Patient consent for publication** Not required.

**Ethics approval** This study involves human participants. Ethical approval was granted by South East Scotland NHS Research Ethics Committee on 28 October 2019, reference: 19/SS/0111. Participants gave informed consent to participate in the study before taking part.

**Provenance and peer review** Not commissioned; externally peer reviewed.

**Data availability statement** Due to the sensitive nature of the data and small sample size, individual patient level data are not available publicly.

**ORCID iDs**
John Joseph Reynolds-Wright http://orcid.org/0000-0001-6597-1666
John Norrie http://orcid.org/0000-0001-9823-9252
Sharon Tracey Cameron http://orcid.org/0000-0002-1168-2276

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
