## [Reviewer comments · BMJ Open]

ARTICLE DETAILS

TITLE (PROVISIONAL)	Using Telemedicine to improve early medical Abortion at Home (UTAH): a randomised controlled trial to compare telemedicine with in-person consultation for early medical abortion
AUTHORS	Reynolds-Wright, John; Norrie, John; Cameron, Sharon

VERSION 1 – REVIEW

REVIEWER	Mazza, Danielle Monash University, General Practice
REVIEW RETURNED	11-Apr-2023

GENERAL COMMENTS	Thanks for the opportunity to review this paper. It is a shame that the trial could not be concluded due to covid and subsequent local policy changes. It means that the trial has limited implications as the only significant finding is related to increased clinical contact time being recorded in the telemedicine group. Even this however is partly explained by the time taken to have the ultrasound as pointed out by the authors in the article. I would suggest changing this article to a letter format that emphasises this latter point In addition the following needs to be addressed: • As this is being published in an international journal it would be helpful to explain what the Scottish public consultation on telemedicine for abortion was and why it impacted on trial cessation• What is SIMD? Please spell this out. If this is a measure of socio economic disadvantage it would be good to have some• The discussion:o has little reference to existing literature on telemedicine abortion and this could be strengthened.o should start out with the main findings of the study instead of the sentence “ The study was designed in line with the CONSORT and MARE guidance(16).• The majority of all other studies of telemedicine in abortion care have been observational cohort studies and so this trial makes a useful addition to the literature.” Should be in the strengths and limitations section of the study• Is there a reference for the following statements?o Models of care where an ultrasound is not required at all will likely reduce overall time in clinical care even further.o but appears lower than reported elsewhere in the literature.
---

REVIEWER	Norman, Wendy The University of British Columbia, Dept of Family Practice
REVIEW RETURNED	13-Apr-2023

GENERAL COMMENTS	Thank you for the opportunity to review this interesting RCT for which I provided review of the protocol in 2020. Most unfortunately, the planned RCT which began recruiting in January 2020 was not completed due to ethical safety concerns related to the pandemic. The authors state the study closed recruiting in August 2020 recruiting about 10% of the required sample size. The title infers that this paper provides the results of an RCT, which may be misleading to readers. The study is not powered to answer the study question, and I think would be better termed a pilot or feasibility study. Further, I note that the authors may have understood that they would be unable to answer the study question prior to submitting the protocol manuscript, as I was asked to review the protocol in the fall of 2020, after recruitment had closed. The study sample of 10% of that required to report a significant "non-inferiority" result is not sufficient to allow the authors to claim to have 'answered' the question of non-inferiority. I would recommend the authors rework the title of this paper, and do a complete overhaul of all sections to make it abundantly clear that this is not a report of a trial, and that sufficient data to draw any conclusions other than for feasibility and acceptability is not available from that collected. Further, the entire results and discussion section must be re-written to remove phrases such as: "Telemedicine for medical abortion appeared to be effective, safe, and acceptable to women but with less time spent in the clinic." Following such an unfounded statement on effectiveness and safety, with further comments that caution related to interpretation due to small sample size, is disingenuous and could be interpreted as overstating results and/or misleading the reader. I am not a statistician and suggest that important suggestions to improve the presentation of results may be provided from someone with statistical expertise.
---

REVIEWER	Weinryb , Maja Karolinska Institute, Women's and Children's Health
REVIEW RETURNED	10-May-2023

GENERAL COMMENTS	Is any data on adherence to treatment available? In the questionnaire supplements I didn't see questions questions about how medication was taken and when (MARE item "13a-3 Include a description of the number of women who used the drug(s) as planned in the protocol (treatment adherence) as well as 13a-4 When more than one drug is used (e.g. mifepristone and a prostaglandin analog), the actual time interval between the agents should be reported, preferably in hours.") Both groups received written information in the packet. In supplement figure 1 this is described as a detailed step-by-step information leaflet. Except instructions, did the written information also provide information about common side effects and guidance on when to contact health care in case of side effects or insufficient treatment effect?
---

VERSION 1 – AUTHOR RESPONSE

Prof. Danielle Mazza, Monash University

Comments to the Author:

Thanks for the opportunity to review this paper. It is a shame that the trial could not be concluded due to covid and subsequent local policy changes. It means that the trial has limited implications as the only significant finding is related to increased clinical contact time being recorded in the telemedicine group. Even this however is partly explained by the time taken to have the ultrasound as pointed out by the authors in the article.

I would suggest changing this article to a letter format that emphasises this latter point
Many thanks for your suggestion, but in line with BMJ Open article guidelines and advice from the Editors we are not able to make this change to the article type.

In addition the following needs to be addressed:

- As this is being published in an international journal it would be helpful to explain what the Scottish public consultation on telemedicine for abortion was and why it impacted on trial cessation
Further detail has been added to this section: The study opened to recruitment on 13 January 2020 and was suspended on 31 March 2020 in response to COVID-19 pandemic, as telemedicine became standard care. The Scottish Government subsequently conducted a public consultation to determine whether there was public support for making the COVID-19 legislative changes permanent (18). The public consultation concluded that the legislative changes allowing telemedicine should continue. This meant that there would not be a return to the previous in-person model of care that the UTAH study was designed around and so the study was formally closed on 31 August 2021.

- What is SIMD? Please spell this out. If this is a measure of socio economic disadvantage it would be good to have some
This is already given in full in the methods section ('study groups' subsection) with a reference: These questionnaire data were combined with routinely collected clinical data on demographics, gestation as determined by pre abortion ultrasound reproductive history and deprivation score (Scottish Index of Multiple Deprivation, SIMD (15)). See supplemental figure 2 for overview of study flow.

- The discussion:

- o has little reference to existing literature on telemedicine abortion and this could be strengthened.

Additional references added to link to statements in discussion

- o should start out with the main findings of the study instead of the sentence " The study was designed in line with the CONSORT and MARE guidance(16).

Section has been restructured

- The majority of all other studies of telemedicine in abortion care have been observational cohort studies and so this trial makes a useful addition to the literature." Should be in the strengths and limitations section of the study

Section has been restructured

- Is there a reference for the following statements?

- o Models of care where an ultrasound is not required at all will likely reduce overall time in clinical care even further.

Sentence modified

- o but appears lower than reported elsewhere in the literature.

Reference added

Reviewer: 2

Dr. Wendy Norman, The University of British Columbia, London School of Hygiene and Tropical Medicine Faculty of Public Health and Policy

Comments to the Author:

Thank you for the opportunity to review this interesting RCT for which I provided review of the protocol in 2020.

Most unfortunately, the planned RCT which began recruiting in January 2020 was not completed due to ethical safety concerns related to the pandemic. The authors state the study closed recruiting in August 2020 recruiting about 10% of the required sample size. The title infers that this paper provides the results of an RCT, which may be misleading to readers. The study is not powered to answer the study question, and I think would be better termed a pilot or feasibility study.

Please see advice from the editor – we are unable to make these changes, rather continue to report as a trial terminated prematurely.

Further, I note that the authors may have understood that they would be unable to answer the study question prior to submitting the protocol manuscript, as I was asked to review the protocol in the fall of 2020, after recruitment had closed. The study sample of 10% of that required to report a significant “non-inferiority” result is not sufficient to allow the authors to claim to have ‘answered’ the question of non-inferiority.

I would recommend the authors rework the title of this paper, and do a complete overhaul of all sections to make it abundantly clear that this is not a report of a trial, and that sufficient data to draw any conclusions other than for feasibility and acceptability is not available from that collected.

Please see advice from the editor – we are unable to make these changes, rather continue to report as a trial terminated prematurely.

Further, the entire results and discussion section must be re-written to remove phrases such as: “Telemedicine for medical abortion appeared to be effective, safe, and acceptable to women but with less time spent in the clinic.” Following such an unfounded statement on effectiveness and safety, with further comments that caution related to interpretation due to small sample size, is disingenuous and could be interpreted as overstating results and/or misleading the reader.

The results and discuss have been edited. The conclusion statement has been edited to match the abstract as per the editor’s remarks as this included appropriate caveats.

I am not a statistician and suggest that important suggestions to improve the presentation of results may be provided from someone with statistical expertise.

Reviewer: 3

Dr. Maja Weinryb , Karolinska Institute

Comments to the Author:

Is any data on adherence to treatment available? In the questionnaire supplements I didn't see questions about how medication was taken and when (MARE item "13a-3 Include a description of the number of women who used the drug(s) as planned in the protocol (treatment adherence) as well as 13a-4 When more than one drug is used (e.g. mifepristone and a prostaglandin analog), the actual time interval between the agents should be reported, preferably in hours.")

These data were not collected in this detail by the questionnaire. But all participants received mifepristone as this had to be administered by a clinician at the time of the study conduct. And when contacting the participants, research staff asked at the start of the contact if they had used their misoprostol as directed and this was confirmed by all participants responding (and this interval was 24-48 hours). A sentence has been added to the results indicating this.

Both groups received written information in the packet. In supplement figure 1 this is described as a detailed step-by-step information leaflet. Except instructions, did the written information also provide information about common side effects and guidance on when to contact health care in case of side effects or insufficient treatment effect?

The written information also included information on what to expect in terms of bleeding and pain. A sentence has been added to the methods section to clarify this.

VERSION 2 – REVIEW

REVIEWER	Weinryb , Maja Karolinska Institute, Women's and Children's Health
REVIEW RETURNED	09-Aug-2023

GENERAL COMMENTS	Thank you for the opportunity to review this manuscript. The clarifications on limitations due to the early closing of the trial, leading to underpowered results has been clarified throughout the manuscript including discussion and conclusions. Thank you for the clarification of what was included in the information pack and how medication was taken. My remaining questions are with regards to the unscheduled clinical contact and time spent in clinical contact. What did unscheduled clinical contact entail – does this include participant initiated contact as well as additional follow ups scheduled after invitation from study staff at 2w follow up (as explained in the first section of the outcomes section)? Does unscheduled clinical refer solely to the follow up appointments for secondary ultrasound after positive LSUPT or were there other forms of unscheduled clinical contact, such as participants reaching out to ask about how to take the medications or if their bleeding was enough/excessive, or inquiring about signs of complications such as infections? Was there any difference in the amount of written information on what to expect in terms of bleeding and when to reach out to health care in case of complications between the groups? These two aspects come to mind since unscheduled clinical contact seems to have been more frequent in the telemedicine group (noting that the sample is small and results are underpowered). As discussed, “If there was a true difference, this may possibly have resulted from greater familiarity and confidence in contacting the clinic via telephone as this was the main method these participants had used prior to abortion”. This makes sense in the case of participant initiated contact, but reading the manuscript it is not clear to me is if contact was initiated by study staff or participants, or both. discussion section:
--

	Post abortion use of effective contraceptive methods were low in both groups. In settings and study groups the respective consultations are described, there is no mention of the contraceptive counselling being carried out at the time of consultation, although this is inferred by the information of which methods could be provided in the medicine pack if desired. Was contraceptive counseling included in the consultations in person/by phone? If so it may be worth clarifying in the settings section describing the standard procedures. Outcomes: total time spent in clinical contact is the statistically significant finding. This measure is defined in outcomes as time to receive EMA, this may be clarified in the outcomes in the abstract as well.
--	---

VERSION 2 – AUTHOR RESPONSE

1) My remaining questions are with regards to the unscheduled clinical contact and time spent in clinical contact. What did unscheduled clinical contact entail – does this include participant initiated contact as well as additional follow ups scheduled after invitation from study staff at 2w follow up (as explained in the first section of the outcomes section)? Does unscheduled clinical refer solely to the follow up appointments for secondary ultrasound after positive LSUPT or were there other forms of unscheduled clinical contact, such as participants reaching out to ask about how to take the medications or if their bleeding was enough/excessive, or inquiring about signs of complications such as infections?

1a) Thank you for highlighting this - this was patient initiated and emergency contact - this has been clarified within the text under outcomes.

2) Was there any difference in the amount of written information on what to expect in terms of bleeding and when to reach out to health care in case of complications between the groups?

2a) This was the same in both groups and has now been emphasised in the 'setting' section

3) These two aspects come to mind since unscheduled clinical contact seems to have been more frequent in the telemedicine group (noting that the sample is small and results are underpowered). As discussed, "If there was a true difference, this may possibly have resulted from greater familiarity and confidence in contacting the clinic via telephone as this was the main method these participants had used prior to abortion". This makes sense in the case of participant initiated contact, but reading the manuscript it is not clear to me is if contact was initiated by study staff or participants, or both.

3a) As above, this has now had further information added to the outcomes section

4) Discussion section:

Post abortion use of effective contraceptive methods were low in both groups. In settings and study groups the respective consultations are described, there is no mention of the contraceptive counselling being carried out at the time of consultation, although this is inferred by the information of which methods could be provided in the medicine pack if desired. Was contraceptive counseling included in the consultations in person/by phone? If so it may be worth clarifying in the settings section describing the standard procedures.

4a) Thank you this has now been clarified in the setting and study groups section - post-abortion

contraception counselling was offered as part of routine clinical care in both groups.

5) Outcomes: total time spent in clinical contact is the statistically significant finding. This measure is defined in outcomes as time to receive EMA, this may be clarified in the outcomes in the abstract as well.

5a) We are now over the word count for the abstract due to formatting requirements and so cannot add any further detail to the abstract, but as open access, I hope that readers will be able to gather this information from the main text.